# Comparative Molecular Dynamics Investigation of the Electromotile Hearing Protein Prestin

**DOI:** 10.3390/ijms22158318

**Published:** 2021-08-02

**Authors:** Gianfranco Abrusci, Thomas Tarenzi, Mattia Sturlese, Gabriele Giachin, Roberto Battistutta, Gianluca Lattanzi

**Affiliations:** 1Department of Physics, University of Trento, Via Sommarive 14, 38123 Trento, Italy; gianfranco.abrusci@alumni.unitn.it (G.A.); thomas.tarenzi@unitn.it (T.T.); 2Cineca, Via Magnanelli 6/3, Casalecchio di Reno, 40033 Bologna, Italy; 3TIFPA-INFN, Via Sommarive 14, 38123 Trento, Italy; 4Department of Pharmaceutical and Pharmacological Sciences, University of Padova, Via Marzolo 5, 35131 Padova, Italy; mattia.sturlese@unipd.it; 5Department of Chemical Sciences, University of Padova, Via Marzolo 1, 35131 Padova, Italy; gabriele.giachin@unipd.it (G.G.); roberto.battistutta@unipd.it (R.B.)

**Keywords:** molecular dynamics simulations, SLC transporters, prestin, Non Linear Capacitance (NLC)

## Abstract

The mammalian protein prestin is expressed in the lateral membrane wall of the cochlear hair outer cells and is responsible for the electromotile response of the basolateral membrane, following hyperpolarisation or depolarisation of the cells. Its impairment marks the onset of severe diseases, like non-syndromic deafness. Several studies have pointed out possible key roles of residues located in the Transmembrane Domain (TMD) that differentiate mammalian prestins as incomplete transporters from the other proteins belonging to the same solute-carrier (SLC) superfamily, which are classified as complete transporters. Here, we exploit the homology of a prototypical incomplete transporter (rat prestin, rPres) and a complete transporter (zebrafish prestin, zPres) with target structures in the outward open and inward open conformations. The resulting models are then embedded in a model membrane and investigated via a rigorous molecular dynamics simulation protocol. The resulting trajectories are analyzed to obtain quantitative descriptors of the equilibration phase and to assess a structural comparison between proteins in different states, and between different proteins in the same state. Our study clearly identifies a network of key residues at the interface between the gate and the core domains of prestin that might be responsible for the conformational change observed in complete transporters and hindered in incomplete transporters. In addition, we study the pathway of Cl${}^{-}$ ions in the presence of an applied electric field towards their putative binding site in the gate domain. Based on our simulations, we propose a tilt and shift mechanism of the helices surrounding the ion binding cavity as the working principle of the reported conformational changes in complete transporters.

## 1. Introduction

Proteins in the solute-carrier (SLC) superfamily are active secondary transporters, whose study has been an active field of research in the last two decades [1,2,3]. Members of the SLC superfamily, which is the second largest family of membrane proteins, play crucial roles in a large number of physiological processes [4]; these range from the transport of amino acids through the cell membrane (SLC1 family [5]), to the regulation of the extracellular concentration of neurotransmitters during synaptic activity (SLC6 family [6]), to the pH regulation of blood [7]. The latter is the case of the SLC4A1 protein, also known as band 3, which transports bicarbonate ions through the plasma membrane of erythrocytes in an electroneutral exchange with chloride ions, a process fundamental for respiration [7]. Among the SLC protein families, of particular interest is the SLC26 one, whose members act as transporters of a broad range of substrates, including Cl${}^{-}$, HCO${}_{3}^{-}$, sulfate, oxalate, I${}^{-}$, and formate [8,9,10,11,12].

Mammalian prestin is an atypical member of the SLC26 family, since it shows no evidence of the transport of ligands across the membrane; instead, it works as a motor protein, and as such it has been extensively studied in the last decade [13,14,15,16,17,18,19]. Prestin consists of 744 amino acids, which can be divided in three major domains: the N-terminus (75 residues), the transmembrane sulfate transporter (TMD) domain (430 residues) and the anti-sigma factor antagonist (STAS) domain at the C-terminus (310 residues); both N- and C-termini are located in the cytoplasm. As a result of binding of an intracellular anion (chloride) to the TMD, the protein undergoes a conformational change that is sufficient to produce a significant increase of its size in the plane of the membrane. This electromechanical feedback mechanism, called *electromotility*, was discovered by Brownwell et al. in 1985 [20].

In humans, prestin is expressed in the lateral membrane wall of specialized auditory sensory cells, namely the cochlear outer hair cells (OHCs) [21,22]; the latter are found in the organ of Corti, the receptor organ of hearing, located in the inner ear. Here, prestin is densely packed in arrays, which elongate and contract in response to the hyperpolarisation and depolarisation of the surface of the membrane [23]. These elongations and contractions have a significant impact on the area of the basolateral membrane, with associated changes in surface area up to 4%; they occur in human cells at frequencies greater than 20 kHz [24] and in other mammalian species even faster, as in the case of the guinea pig’s OHCs, with changes occurring 120 microseconds after stimulation [25]. Such conformational transitions, originating electromotility, are responsible for the amplification of sound; mutations in human prestin impairing electromotility seem to lead to severe diseases, like non-syndromic deafness [26].

Electromotility of mammalian prestin is associated to a nonlinear dislocation of charges across the cell membrane, following stimulation from an external voltage—a phenomenon known as Non Linear Capacitance (NLC) [27]. This movement of charges can be explained by the relative motion of a charged moiety of the protein in the OHC lateral membrane or by a partial translocation of the ligand, namely the chloride ion. On the opposite, the membrane responds as a classical linear capacitor in the presence of non-mammalian ortholgs of prestin, which adhere to the function of the SLC26 family as antiporters, with bicarbonate ions being exchanged for chloride ions. These different mechanisms of actions are represented in Figure 1. In response to an applied voltage, the membrane would exhibit NLC when containing a mammalian protein such as rat prestin, or an electric current when containing non-mammalian orthologs [14].

Numerous experimental studies have been dedicated to prestin and its orthologs. Some of them focused on the role of anions, in particular chloride, to trigger NLC [13,14]; other studies investigated how various mutations affect the electromotily and the transport of charges in the family members [17]. However, how prestin is able to give rise to the OHC elettromotility process is still unclear. The formation of oligomers (dimers or dimers of dimers) in the basolateral membrane seems important and, in this process, the STAS domain plays a fundamental role, as demonstrated by the structures of dimeric SLC26A9 [28]. However, the TMD domain of the single protomer is the basic unit for the translocation of anions [18]; since this (incomplete) ion transport is at the origin of NLC in prestin, we focused here on the TMD domain of the single protomer.

Specifically, in order to investigate the different behaviour of prestin when expressed in mammals, or in one of its non-mammalian orthologs, we present here a computational comparative study of TMD from rat and zebrafish prestins (rPres and zPres, respectively), in two conformations: an inward open, with the putative binding site accessible to the solvent from the cytoplasmic side of the cell, and an outward open conformation, where the binding region is accessible from the exterior of the cell. Although the latter arrangement is not expected to be biologically relevant for the mammalian ortholog, its structural investigation is nonetheless informative on the differences between the two protein systems. Through our bioinformatics analyses and molecular dynamics (MD) simulations we aim at pointing out the interactions stabilizing the inward/outward open conformations, and identifying the residues that define the binding site of chloride ions. The interaction of the latter with the protein is enhanced, in a subset of simulations, through the application of a transmembrane electric field. Results from the computational investigation are compared with experimental observations available in literature: in particular, structural data from homologous proteins and experimental measures of water accessibility are employed to validate the initial protein models, while data from mutagenesis experiments, accompanied by measures of NLC, are used to confirm the functional role of specific residues, emerging from the simulation study.

## 2. Methods

### 2.1. System Setup

Given the lack of experimentally solved structures, the initial models of rPres and zPres, in inward-open (rI and zI, respectively) and outward-open (rO and zO, respectively) conformations, were obtained through homology modelling. In previous studies, different transporters were used as templates: Lovas et al. [29] employed the glutamate transporter in a folded conformation consisting of 8 transmembrane helices, while Gorbunov et al. [18] proposed an experimentally-validated 14TM model, lately confirmed by the experimental structures of the related SLC26Dg [30] and SLC26A9 [9]. Hence, the 14TM models were developed using as templates the transporter SLC26Dg (PBD ID: 5DA0) [30] for the inward open conformations, and the Band 3 transporter (PDB ID: 4YZF) [7]—belonging to the SLC4 family—for the outward open ones since there are no experimental structures for the outward-open conformation of proteins from the SLC26 family. The sequences were aligned by hidden Markov models (HMMs) profile comparison through the Phyre2 server [31]. The 3D coordinates were obtained using the homology modeling module of the MOE suite [32] by generating 10 models which were subjected to an energy minimization (RMS gradient 0.5 Å, force field AMBER12:EHT) and finally scored by GB/VI method. The model with the lowest GB/VI energy was treated by the Protonate3D tool [33] to set a suitable protonation state of ionizable sidechains and used for MD studies.

Each system was embedded in a lipidic membrane of 1-palmitoyl,2-oleoyl-sn-glycero-3-phosphocholine (POPC) of 110 × 110 Å${}^{2}$ surface area in the xy-plane, built with the plugin Membrane Builder of the VMD software package [34]. Phosphatidylcholine was chosen since it represents one of the the most abundant components of eukaryotic cell membranes [35] and is widely employed in MD simulations of membrane proteins [36]. The size of the patch ensured that the protein had a minimum distance of 18 Å from its own images, under periodic boundary conditions. The OPM web server [37] was employed to obtain the proper orientation of the proteins inside the bilayer. Overlapping lipids within 0.8 Å of the heavy atoms of the protein were removed from the membrane patch to allow protein insertion without steric clashes. The system was finally solvated with TIP3P water molecules [38]. Na${}^{+}$ and Cl${}^{-}$ ions were added to neutralise the system and reach the physiological concentration (0.15 M). The VMD plugins Solvate and Autoionize were employed to accomplish these tasks. The four setups consisted of ≈110,000 atoms each, and the unit cell had initial dimensions of 110 × 110 × 105 Å${}^{3}$. As an example of the final setup, the membrane-embedded rI is shown in Appendix A.

### 2.2. Simulation Details

All simulations were performed using the software package NAMD 2.12 [39] with the force field CHARMM36m [40]. A cutoff of 12 Å was used for the short-range components of the non-bonded interactions. The long-range electrostatic calculations employed the Particle Mesh Ewald [41] method with a grid spacing of 1 Å. Periodic boundary conditions were used along the three axis.

All systems were minimised for 2000 steps to allow the proteins to relax and to remove steric clashes. The equilibration phase was performed in three steps. First, a 500 ps run in the NVT ensemble was performed, using a time step of 1 fs and a temperature of 310 K. All atoms were kept fixed with position constraints, except for the lipidic tails of the POPC molecules. This procedure allowed melting of the lipids, thus reducing the gap between the protein and the membrane. A second run of 500 ps in NVT was performed with a 2 fs integration step. Protein and water molecules in its cavity were kept fixed, with the other components free to move. Custom forces were applied to push water molecules out of the hydrophobic region of the system, and thus accelerate the adhesion of the lipids onto the transmembrane surface of the protein. The last step of equilibration was performed in the NpT ensemble, at a pressure of 1 atm and with oscillations along the *z*-axis decoupled from those in the xy-plane. Protein atoms were harmonically restrained to their initial positions and custom forces were applied on water molecules still trapped in the region between the protein and the membrane. A simulation time of 1500 ps was sufficient to remove all water molecules in the lipidic region, and close the gap between the protein and the membrane.

The last frames of the previous step were used as input structures for the production runs in the NpT ensemble. The temperature was kept at 310 K [42] with a damping factor of 1 ps${}^{-1}$. Constant pressure was achieved by the Nosé–Hoover Langevin barostat [43,44] with 1 atm of target pressure, 200 fs oscillation period and 100 fs decay coefficient. Bonds between heavy atoms and hydrogen atoms were constrained with the SHAKE [45] algorithm and the coordinates were recorded every 4 ps. Each system was simulated for 700 ns on the CINECA supercomputer Marconi.

A second round of 200-ns simulations was performed for the inward structures in the presence of a constant external electric field in the *z*-direction, to mimic the presence of a cross-membrane voltage of 100 mV.

### 2.3. Analysis

Multidimensional scaling (MDS) method was used to project the conformations of the proteins on two dimensions [46]. Each frame corresponds to a point, whose arrangement with respect to the others is defined by the matrix of root-mean-square deviations between conformations, used as a distance matrix. The relative distances between points are preserved. Given the lack of knowledge about the number of clusters and distribution of the data, the Gaussian Mixture model (GMM) was applied to define in a probabilistic manner how the conformations can be partitioned in groups [47]. This model assumes that the data are distributed according to a fixed number of multidimensional gaussians. The algorithm assigns iteratively a probability of membership for each point in each cluster and updates for each cluster the parameters of the gaussians, until convergence. To select the optimal number of clusters, the Bayesian Information Criterion (BIC) was used [48]. Increasing the number of gaussians introduces new parameters in the model that can allow a better fit; however, to avoid overfitting, the BIC introduces a penalty based on the number of parameters, and the model that returns the smallest BIC is chosen as the best clustering result.

PyInteraph analysis tools [49] were applied to the four trajectories in order to identify inter-residues communication pathways within the simulated systems. The core idea is that a protein can be represented as an undirected graph. The nodes represent the amino acids of the chain and the edges are defined according to the interactions between residues in the structural ensemble. From the network it is then possible to investigate long-distance communications within the protein and the role of individual residues within the framework of the protein function. In order to apply the tool to membrane proteins, the algorithm was converted to python 3, introducing flexibility in the definition of the interactions. The persistence thresholds were chosen according to the size of the largest hydrophobic cluster criterion [50], obtaining threshold values ranging from 16.0 for zI to 28.0 for rO.

Principal component analysis (PCA) was performed on all simulations of the inward-open states, for both rPres and zPres, in order to compare the explorations of the conformational space in presence and in absence of the electric field. The reference structures for the PCA were obtained by averaging over all the conformations in the production simulations, after alignment to the last frames. The extraction of the essential subspace was performed using MDAnalysis [51].

The calculation of the displacements of the TM center of mass along the *z*-axis was performed with the Gromacs tool gmx traj, and the calculation of the helix tilts with the tool gmx bundle [52].

## 3. Results and Discussion

### 3.1. Structural Models and Validation

TMD ranges from residue 75 to 504 in rPres and from residue 76 to 507 in zPres; in this region, the two proteins share 63.1% of sequence identity and 85.8% of sequence similarity. The alignment of rPres and zPres sequences is shown in Figure 2. Thorough analyses on the conservativity of prestin residues are already present in the literature, both as comparisons among sequences of SLC26 homologues (including prestin) [28] and among mammalian and non mammalian prestin sequences (including rat and zebrafish ones) [53]; the conservativity of rPres/zPres residues will be thus referred to these studies throughout this work.

Given the lack of experimentally solved structures of prestin proteins, simulations were started from homology models of the TMD for both the rat (r) and zebrafish (z) prestins orthologs, in the inward-open (I) and outward-open (O) conformations (Figure 3): specifically, SLC26Dg was used as a template for the I states, while the Band 3 transporter was used for the O states. Despite the lower sequence identity between the TMD of rPres and SLC26Dg (23.3%) with respect to murine and human SLC26A9 (40.3% and 40.5%, respectively), whose structures in the inward-open state have been recently solved (PDB ID 6RTC [9] and 7CH1 [28]), SLC26Dg was chosen because of the large amount of experimental data available in literature. Specifically, in the case of SLC26Dg, most of these experimental data focus on the structural features and the functional role of the transmembrane domain as a transporter [30], which is the core of the current study; on the other hand, more recent insight on the SLC26A9 protein focused on the dimerization process and on the role played by the STAS domain and the N-terminal [28]. In addition, it was shown that SLC26Dg functions as a typical transporter, while there is no clear evidence on whether SLC26A9 behaves like a channel or a transporter [28]. For these reasons, we considered SLC26Dg as a better template for prestin from the functional point of view. The models of rI and zI aligned to the SLC26A9 structure show nonetheless a good agreement (Section S1.1). In addition, our rI model was validated against published data of water accessibility in mammalian prestin [18], showing excellent agreement between the experimental results and the exposure of residues to solvent as computed from our MD simulation (Section S1.2).

In our models, the TMD consists of 14 transmembrane helices (TM) arranged in two groups, TM 1-7 and TM 8-14, pseudosymmetric to each other (Figure 4). α-helices represent the majority of secondary structure elements, except for the beginning of TM3 and the terminal part of TM10, that contain antiparallel β-strands. In the three-dimensional arrangement, the TM helices are intertwined and organised in two bundles, namely the core and the gate domains (Figure 4), according to the nomenclature introduced for uracil permease (UraA) [55]. The core is a convex structure, comprising TM1-4 and TM8-11, while the gate domain, consisting of the remaining helices TM5-7 and TM12-14, has a concave shape. The putative binding site of the chloride ion is located in the core domain, between TM3 and TM10 [18]; the binding pocket was suggested to be delimited on one side by the two antiparallel β-strands, and on the other by the gate interface. Data available on the SLC26 family, supported by those on members of SLC4 and SLC23 families, strongly suggest an elevator-like mechanism of translocation of the bound ions in zPres [9,12,56]; following a vertical translation of some elements of the core relative to the gate domain, the interface between the gate and the core opens toward the intracellular or the extracellular side, allowing the translocation of the bound ions.

### 3.2. Conformational Variability

Equilibration of the starting models was assessed through the calculation of root-mean-square deviation (RMSD) matrices, computed on C${}^{\alpha }$ atoms (Section S2). The converged portions of the trajectories were determined after filtering out the large fluctuation of the extracellular loops, quantified by the root-mean-square fluctuations (RMSF) of C${}^{\alpha }$ positions (Appendix A). In the simulations of rI and zI, the equilibration time was taken as the first 50 ns and 100 ns, respectively. In the outward open conformations, both protein models require longer equilibration time, and the analysis is performed on the last 350 ns of the trajectories. The multidimensional scaling (MDS) method [46] was employed to project the sampled conformations on two dimensions, with the aim of facilitating the investigation of the conformational space explored during the simulations. The RMSD matrices (after RMSF-based filtering) were used here as a measure of distance between the structures. Appendix A shows for each system a non-uniform distribution of conformations, resulting in a number of conformational clusters that ranges from 4 in the case of rI to 7 in the case of zI.

The analyses of secondary structure along the trajectories reveal a high stability of the transmembrane helices, whose structure is preserved in all the systems (Appendix A and Appendix A); an exception is given by the short TM10, facing the cytoplasmic side, which appears to be less stable in zI than in the other systems. As expected, large variability is observed in the intra- and extracellular loops connecting the helices, which transiently adopt or lose ordered conformations.

A first comparison of the starting models with the equilibrated structures discloses a tendency of inward structures to preserve the arrangement of the helices with only minor changes (Figure 3). On the other hand, the outward-open states require significantly longer equilibration times, resulting in larger rearrangements of the starting models. Specifically, in the outward states there is a reduction of the proteins’ size upon equilibration, mostly on the extracellular side. In zO, the largest rearrangement involves TM7, TM5 and the extracellular N-ter part of TM4. The latter moves toward the C-ter of TM5 (Appendix A); this concerted movement brings together two regions of the gate and the core located at the opposite sides of the structure, thus reducing the size of the upper region of the protein. rO undergoes different rearrangements during equilibration, more distributed throughout the structure, and with smaller displacements with respect to zO. The largest deviation is observed in TM6; the extracellular loop between TM5 and TM6 tends to move towards the gate, but the displacement of TM4 is less pronounced (Appendix A). These rearrangements result in some discrepancies between the equilibrated rPres and zPres outward-open conformations, particularly in the disposition of TM6, TM7, and TM14 of the gate domain, which determines a larger cross-sectional area in the membrane plane for rPres. Smaller deviations are observed in the core, namely at the extracellular end of TM4 and at the intracellular end of TM2, which is closer to the core/gate interface in zO.

To identify putative conformational changes associated with the transporter activity, we performed a structural alignment between the conformations sampled from the rO and zO simulations and the most representative structures of rI and zI, respectively. Two independent alignments were performed for each ortholog, with respect to the core and the gate domains, respectively. In this way we identified the helices undergoing the largest displacements with respect to the other TMs belonging to the same domain. The RMSD between the simulated outward open conformations and the most representative structure of the inward simulation (Appendix A) displays a higher rigidity in the arrangement of the core helices when transitioning between the the two states, with respect to the gate. Interestingly, this difference is more pronounced in zPres rather than in rPres, whose outward-facing state is not expected to be biologically relevant. Helices undergoing the largest displacement in the gate of rPres are TM6-7 and TM13-14; in the core, particularly large rearrangements with respect to the other helices are found in TM4 (Appendix A and Appendix A). In zPres, the largest rearrangements are still found in TM6-7 and TM13, with higher displacements than rPres (Appendix A and Appendix A). A high mobility of the gate was reported also in previous simulations of the transporter SLC26Dg in monomeric form [12]; in that case, the creation of a large scaffold upon dimerisation was suggested to provide higher stability during the displacement of the core in the full transporter.

Experimental observation on the SLC26A9 transporter in the two conformations suggested that the core domain, where the putative binding site of the ion is located, undergoes an upward rigid-body movement relatively to the gate [9,56,57,58]. In order to assess this in our simulations, we compared the equilibrated inward- and outward-open conformations of the full TMD, upon structural alignment of the sole gate. Starting from the aligned trajectories, we computed the shift of the center of mass of each core TM, along the direction perpendicular to the membrane plane (Figure 5). In the case of rPres, the displacements are distributed in both directions, meaning that some core TMs undergo a movement toward the intracellular side upon transition to the rO structure, contradicting our expectations; this further motivates the non-physiological role of rO. On the other hand, in the case of zPres, there is a clear shift toward the extracellular side for all the core TMs. Particularly large deviations are observed for TM1, TM10 and TM11, which are spatially close to each other in the helix bundle. Such conformational change might be facilitated upon dimer formation, since these helices are located on the opposite side of the protein with respect to the dimer interface [12]; while the latter is stabilized in the membrane, helices located at the opposite side may indeed acquire a larger conformational freedom. In our simulations of zPres, the collective translation of the core domain upon conformational change leads to an upward shift of the beta-sheets hosting the putative Cl${}^{-}$ binding site; such movement is necessary to expose the binding cavity toward the extracellular space in the full transporter. A displacement of the Cl${}^{-}$ binding site is observed also in rPres; however, the entity of the shift is smaller than in the case of zPres.

The translation of the core helices in the direction perpendicular to the membrane is accompanied by a tilt of the individual TMs with respect to the inward-open conformation (Figure 5). In zPres, there is a clear distinction between the behavior of the helices belonging to the inner shell (facing the gate domain) and those in contact with the membrane; the latter are tilted in zO in a direction perpendicular to the bilayer, while the former become more parallel to it. In particular the large tilt of TM10, accompanied by a large upward shift, leads to the observed displacement of the putative binding site toward the extracellular side; at the same time, the codirectional tilts of both TM3 and TM10 prevent the rupture of the β-sheet hosting the binding site.

### 3.3. Key Residues in the Interaction Network

We employed PyInteraph [49] to determine the pathways of communication that extend through non-bonded interactions, with the aim of identifying network hubs and understanding how they affect the conformational changes associated to NLC and the transport of ions. The network of interactions shows that the four systems highly favour hydrophobic contacts; in fact, the residues with the highest numbers of connections are mainly alanine, valine, leucine and isoleucine (Table 1).

Table 1 displays a significant difference between the distributions of hub residues among the gate and the core, in the case of zPres and rPres: in the former, the vast majority of hubs is located in the core (83% in the core and 17% in the gate for zI; 75% in the core and 25% in the gate for zO), while in the latter the number of hubs is balanced between the two domains (58% in the core and 42% in the gate for rI; 43% in the core and 57% in the gate for rO). These results may explain the rigidity of the gate domain in rPres when compared to zPres, as observed in the previous section. In addition, the distribution of these residues on the structure leads to a larger number of hubs at the core/gate interface in rI with respect to zI (Figure 6 and Figure 7), arguably preventing the two domains of rPres to rearrange with respect to each other and thus undergo large conformational changes as in the case of zPres. When comparing the inward- and outward-open conformations, only two residues maintain a high centrality, namely M143 (TM3, core) and A218 (TM5, gate) in the case of rPres, and V108 (TM1-2, core) and L342 (TM8, core) for zPres. In the case of rO, M143 and A218, located in the core and in the gate respectively, are both exposed on the inter-domain contact area; in rI, only M143 faces the interface. In zPres, the two central residues V108 and L342 are buried in the core domain. This results in the absence of common hubs between zI and zO at the core/gate interface, thus allowing the full transporter zPres to undergo larger rearrangements between the two domains from the inward to the outward state and viceversa.

About 50% of the hubs identified in rO correspond to the most central residues of zO (according to the rPres/zPres numbering: V92/V93, and M143/M144, A217/A220, A218/A221). Hubs that are not shared by the two systems are mostly located in the gate domain in the case of rO and in the bulk of the core domain in zO; this may account for the higher rigidity within the core domain observed in zPres, with respect to rPres, as reported in the previous section. Comparison between the rI and zI results reveals three common residues: according to the rPres/zPres indexing, they are L95/L96, A102/A103, V107/V108. These residues are located at the extracellular end of TM1 (C-term). In rI, they form a hydrophobic cluster by interacting with A100, together with the highly central residues L104 (also located in TM1), I443-V444 (located in the gate, in the middle of TM12), L435 (located in the external loop between TM11 and TM12). This hydrophobic cluster closes the upper side of the binding cavity, holding the gate and core together (Appendix A).

The results from the network analysis above strengthen the hypothesis of a functional role for this region in rPres, as already suggested by mutagenesis experiments performed in previous works [15,16], where the mutations of A100 and A102 into leucines or valines were reported to abolish NLC. In our simulations, A100 and A102 form a bridge between the core and gate domains and guide F101, one of the three residues whose simultaneous mutation into the corresponding zPres residues (namely L93M, F101Y, P136T) is sufficient to suppress NLC in rPres [17]. The analysis of the network shows that F101 is in the middle of the shortest path connecting L93 and P136; since the single mutation F101T does not destroy the NLC response, it is likely that A100V/L and A102V/L interrupt the interaction path, and the triplet of residues loses its functional role. While the preservation of the hydrophobic network could be important for stabilizing rPres in the inward-open state, we suggest that its disruption plays a functional role in the conformational change of the transporter upon interaction with Cl${}^{-}$. The different response elicited in rPres and zPres by substrate binding may be attributable to differences in the binding cavity (see below).

The interaction network analysis suggests the presence of another important cluster of contacts centered on TM1, which might play a functional role in the transition between the inward and outward states of zPres. First, Table 1 highlights the presence of several central TM1 residues in zI, while only one is found in zO, suggesting that TM1 might indeed play a role in stabilizing the inward-open state. In addition, Q98, Y102 and M104 on TM1 form stable interactions in zI with N450 (TM12) and Y231 (TM5), which belong to the gate domain. Such contacts stabilize a bent conformation of TM1, which covers the putative binding site from the top, making it inaccessible from the extracellular side (Appendix A). These bonds are broken in zO; as observed previously, TM1 is indeed one of the helices of the core undergoing the largest rearrangement during the conformational transition, after gate alignment. As highlighted in the next sections, Y102 plays a key role in binding the incoming chloride ion in zI; such event might lead to the breaking of H-bonds between the core and the gate, and to the movement of TM1 that accompanies the inward/outward transition. Importantly, Y102 is conserved in non-mammalian prestin, while the same position corresponds to a conserved phenylalanine in mammalian prestin [53]. On the other hand, zO is stabilized by new, different electrostatic interactions between the intracellular sides of the gate and the core, involving K286-E407 (TM7-TM10, respectively), V213-T363 (TM5-TM8), and S90-I293 (TM1-TM7).

A persistent interaction pointed out by the network analysis is the salt bridge between K286 in TM7 and E407 in TM10 of zO, facing the cytoplasmic side of the bundle. These residues do not directly interact in the inward conformation and their distance is ∼28 Å. The situation dramatically changes in the outward conformation, where the two residues form a salt bridge. We notice that this interaction was not captured by the initial homology model, where the two amino acids were not in contact; the interaction is established only after 300 ns of MD simulations (Figure 8). The vicinity of the two helices upon formation of the salt bridge keeps the gate and the core in close contact on the intracellular side; although there is no previous evidence that this salt bridge is directly involved in the transport process, our analysis suggests that it may stabilize the outward structure of the full transporter. This hypothesis is further supported by the comparison with rPres in the outward conformation, where K286 and E407 are conserved at positions 283 and 404, respectively: in this case, the transmembrane helices TM7 and TM10 do not get in contact. Since rat prestin in the outward conformation is not a physiological viable state, the electrostatic interaction might reveal a prominent functional role for the transport activity, that is exclusive of the non-mammalian system.

A close packing of TM6 and TM12 in the gate domain is observed in rI, but missing in rO. These contacts are stabilized by the interaction between L272 of TM6 and I443 of TM12, which corresponds to a highly central residue of the rI protein network. Residue L272 belongs to a short sequence of six residues that are highly conserved among mammalian prestins (sequence GLLLGG, from residue 270 to 275), but are indeed variable in other species. The high degree of conservation in rPres suggests a functional role of this segment in NLC. The abundance of glycines in the conserved sequence might allow enough conformational variability to this region, with the possibility for the three leucine residues to easily form hydrophobic interactions upon conformational changes induced by NLC. As highlighted in the previous section, the TM6 conformation is one of the main differences between rI and rO; in the latter case, the contact between the TM6 conserved sequence and the inner shell is broken, leaving TM6 as one of the main determinants for the extended shape of rO in the plane of the membrane (Appendix A). On the other side, the corresponding sequence in zPres (sequence VFLYII, residues 273–278) includes large hydrophobic residues with a lower degree of flexibility and able to form a large contact surface with the inner TMs—as indeed observed in both the zI and zO trajectories, following equilibration.

Another interesting cluster identified in rI is the one comprising two residues of TM3, namely M143 and V147, in interaction with L488 (TM14) and V182 (TM4), respectively. Both TM4 and TM14 are involved in the formation of the dimer in SLC26A9 [28]; we may speculate that the insertion of ions in the binding pocket close to TM3 might elicit conformational variations that extends to TM4 and TM14, possibly leading to an interplay between the two protomers of the dimeric form of the transporter, in a sort of reciprocal allosteric regulation. This hypothesis, however, requires further experimental validation.

### 3.4. MD Simulations with an Applied Electric Field

Prestin’s activity is modulated by the binding of chloride ions on the intracellular side of the protein; this event is thought to initiate the conformational changes that lead to NLC [15,59]. During the two 700 ns-long simulations of the inward states, no binding events occurred, although Cl${}^{-}$ ions explored the binding cavity in the zebrafish setup. In order to faciltate the interaction between the protein and the ions, we simulated the rI and zI systems for additional 200 ns in the presence of an electric field directed perpendicular to the membrane, with a voltage of 100 mV along the simulation cell [60].

We performed a Principal Component Analysis (PCA) on the equilibrium simulations without the applied electric field. The trajectories with the applied field were then projected on the obtained first three principal components to assess whether the proteins explored the same regions of phase space in the presence of the electric field. The whole analysis was limited to the C${}^{\alpha }$ atoms only. The results show that the electric field does not modify the global dynamics of the inward systems; the assumed conformations overlap with the regions previously explored by the systems (Appendix A and Appendix A).

However, the presence of the electric field enhanced the exploration of the binding cavity by chloride ions in both systems (Appendix A). Interestingly, a different number of binding events is observed in the two cases: only 3 binding events occurred in the simulation of rPres, while 22 binding events were observed in the simulation of zPres. The high number of ions interacting with zPres can be attributed, at least partially, to its larger cavity when compared to the one in rPres. Volumetric analysis of the binding pocket, performed with fpocket [61,62], yields indeed a pocket volume of 0.61±0.01 nm${}^{3}$ in rPres, and a volume of 1.22±0.01 nm${}^{3}$ in zPres. Visual inspection of the trajectory suggests that this feature may be due to the loss of secondary structure in the short α-helix located after residue SER401 (in TM10), as pointed out in Figure 9. In addition, substantial differences in the binding site between rPres and zPres play a crucial role in driving ion permeation, as highlighted below.

The pathways that lead to ion binding are mainly driven by residues in the gate domain; however, the most stable interactions (referred here as "binding events") are mainly formed with residues of the core. This observation is consistent with previous experimental investigations of the residues coordinating the ligand in uracil permease [55]. In rPres, residues R211 in TM5, K276 in TM6, Q454 and K449 in TM12, all belonging to the gate, are the ones establishing the highest number of contacts with the ions approaching the binding pocket; we assume that they serve as anchor points for the ion to remain in the proximity of the protein. In zPres, Cl${}^{-}$ ions make close contacts with R214, K279, Q457 and R459 in the gate domain; however, the majority of anions access the cavity from the cytoplasmic end of TM5 and TM12 of the gate.

After entering the binding pocket of rPres, the most stable interactions found between the ions and the protein involve S396, L397 and S398, located in the core domain. Once the hydrogen-like bond is formed with the sidechains of the serine residues and the backbone nitrogen atom of leucine, the ion persists in the pocket for several nanoseconds. In zPres, ions establish halogen bonds with S399/S401 residues. In both rPres and zPres cases, while the ion is still in the binding pocket, the halogen bonds can break, and the ions move towards nearby residues, namely F101/P136 of rPres and Y102/T137 of zPres (Figure 9). Eventually, the detachment of the ion from the binding site is governed by the rotameric configurations of S398/S401.

The presence of non polar residues, as found in rPres, is a feature frequently found in anion binding sites [63,64] and is in agreement with the putative binding site of SLC26A9 protein [9]. These non polar residues were shown to be fundamental for the NLC in rPres: mutations of L93, F101 and P136 with the corresponding zPres residues (M94, Y102, T137) abolished NLC [17]. F101 and P136, moreover, are fully conserved in mammalian prestin, while the corresponding positions are invariably occupied by a tyrosine (Y102 in zebrafish) and a threonine (T137 in zebrafish) in non-mammalian species; among other human SLC26 transporters, the pair F101/P136 is present only in member A3 [28,53]. On the other hand, the presence of conserved polar moieties in the binding site of zPres and in general in non-mammalian species (specifically, Y102 and T137) facilitates the interaction with the ions, as pointed out by the higher number of binding events found in zPres with respect to rPres. We suggest that this stronger interaction between the anion and the hydroxyl groups of Y102 and T137 is the key initial event that triggers the “opening” of the cluster of interactions located above the binding site (Figure 10), thus initiating the conformational changes that lead to the formation of the outward-open state. In mammalian prestins the presence of a phenylalanine and a proline in the corresponding positions leads to a weaker interaction with chloride anions that remain trapped in the binding site held in the inward conformation, unable to move towards the outward one. These trapped anions are thus sensible to variations in the transmembrane potential and may constitute the voltage sensor for the NLC properties of mammalian prestin.

## 4. Conclusions

We simulated the mammalian motor protein prestin (rPres) and the ortholog, non-mammalian transporter (zPres). For each protein, our simulations were started from two distinct conformations, the inward-open and the outward-open ones, where the ion binding site is exposed towards the intracellular or extracellular side, respectively. Although the outward-open state of rPres is not biologically relevant, the comparison between the four systems allowed us to identify structural features contributing to their working mechanism, with a specific focus on the determinants that make rPres an incomplete transporter. Given the lack of experimental structures, starting configurations were obtained by homology modeling, and the resulting models were validated through comparison with the only available resolved structure of an SLC transporter, namely SLC26A9. Further validation of the rI model was performed through comparison with experimental data of water accessibility.

In rPres, we identified a large number of highly central residues located at the interface between the gate and the core domains. The experimental evidence pointed out a conformational transition taking place in the full transporter as a translation of the core domain with respect to the gate: here, we identify a network of interactions that can hinder this domain translation, thus preventing the conformational change in rPres. Specifically, TM1 is involved in a network of hydrophobic interactions just above the binding site, keeping the core anchored to the gate. At the same time, these interactions are relevant for NLC in rPres, as previously reported following mutation of apolar residues in this region. In zPres, a number of H-bonds between TM1 and the gate stabilizes the inward-open conformation. One of the residues involved, Y102, is a key residue for the interaction with the incoming ion; upon ion binding, the core/gate contacts are likely to be weakened, thus allowing for the transition toward the outward-open state.

We analyzed also the differences in the binding site between rPres and zPres by inspecting the interactions between Cl${}^{-}$ ions and the protein in the presence of an applied electric field, directed along the membrane axis. In this respect, a key role is played by residues L93, F101 (TM1) and P136 (TM3): these three residues face the binding cavity, just above S398, which is relevant to guide the incoming ion within the pocket. In zPres, F101 and P136 are replaced by Y102 and T137; we hypothesise that the interactions between the chloride ion and the hydroxyl groups of Tyr and Thr are the starting event opening the gate located above the binding cavity, as required by the conformational transition of the full transporter.

Our comparative investigation of prestin represents an additional step for the understanding of NLC, although further computational studies are desirable to elucidate the possible cooperative mechanisms upon dimer formation, or the interaction between the TMD and STAS domains. In addition, further experimental evidence may shed more light on the proposed mechanism and the critical assessment of the interaction networks pointed out by our computational analysis; in particular, functional assays conducted on cells expressing prestin mutants, monitored by physiological techniques such as patch-clamp, may contribute to assessing the functional role of the key residues identified in this study.

## Figures and Tables

**Figure 1 ijms-22-08318-f001:**
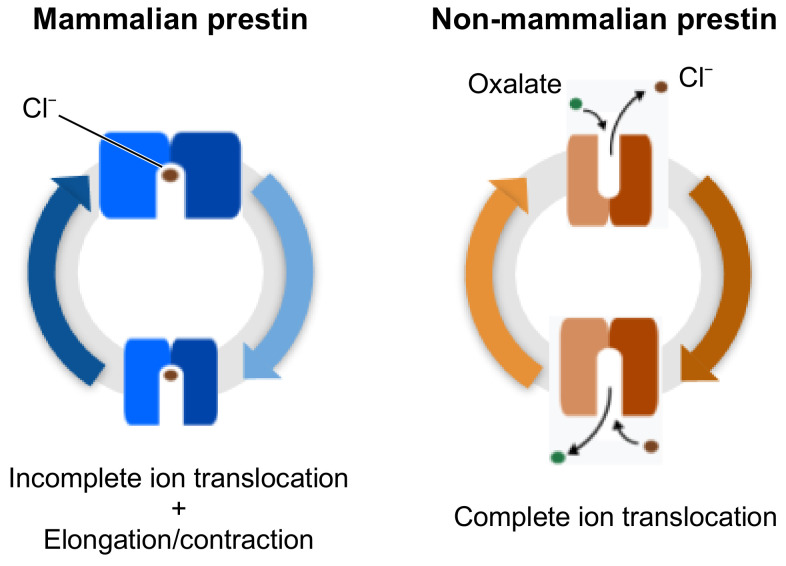
Schematic representations of the electromotile behaviour of prestin in mammals (blue), and of the full substrate transport process in non-mammalian prestin (orange).

**Figure 2 ijms-22-08318-f002:**
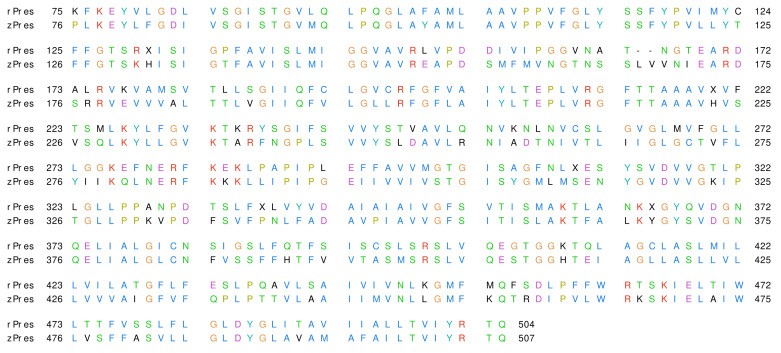
Alignment of rat and zebrafish prestins (rPres and zPres, respectively). The residues are colored according to the ClustalX scheme [54].

**Figure 3 ijms-22-08318-f003:**
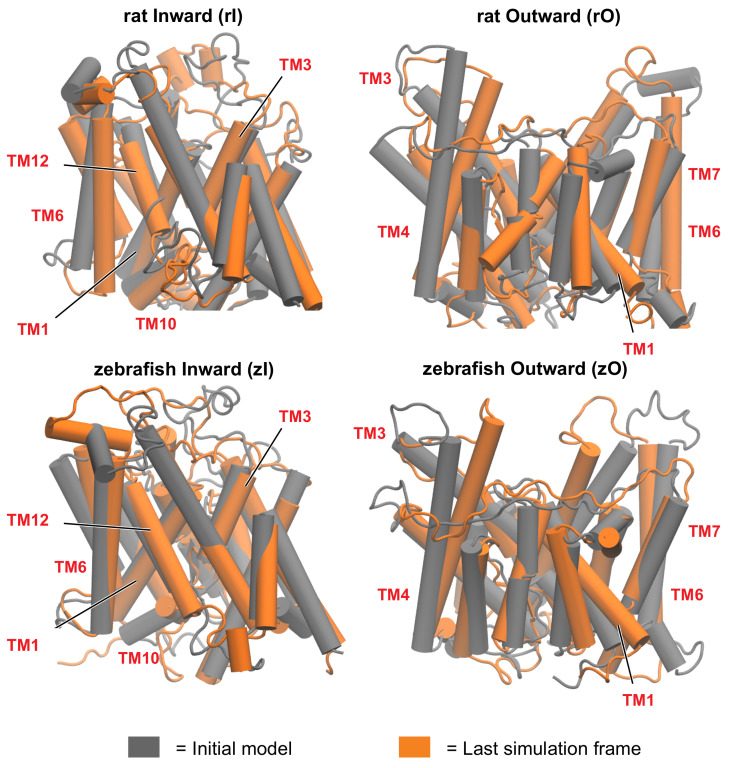
First (grey) and last (orange) frame of the simulation, for rat and zebrafish prestins in the inward-open and outward-open conformations. The rearrangement of the helices is more pronounced in the outward open conformations. Some of the most functionally relevant transmembrane helices (TM), as highlighted in this study, are labelled.

**Figure 4 ijms-22-08318-f004:**
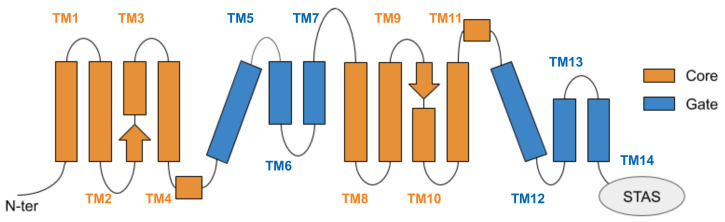
Schematic representation of the transmembrane domain of prestin, located between the N-terminal and the cytoplasmic, anti-sigma factor antagonist (STAS) domain. Helices are arranged according to a 7 transmembrane inverted repeat fold; they are conventionally indicated from 1 to 14, starting from the N-term, and are divided among the gate and core domains. The two arrows represent the short β-strands hosting the putative binding site of chloride ions.

**Figure 5 ijms-22-08318-f005:**
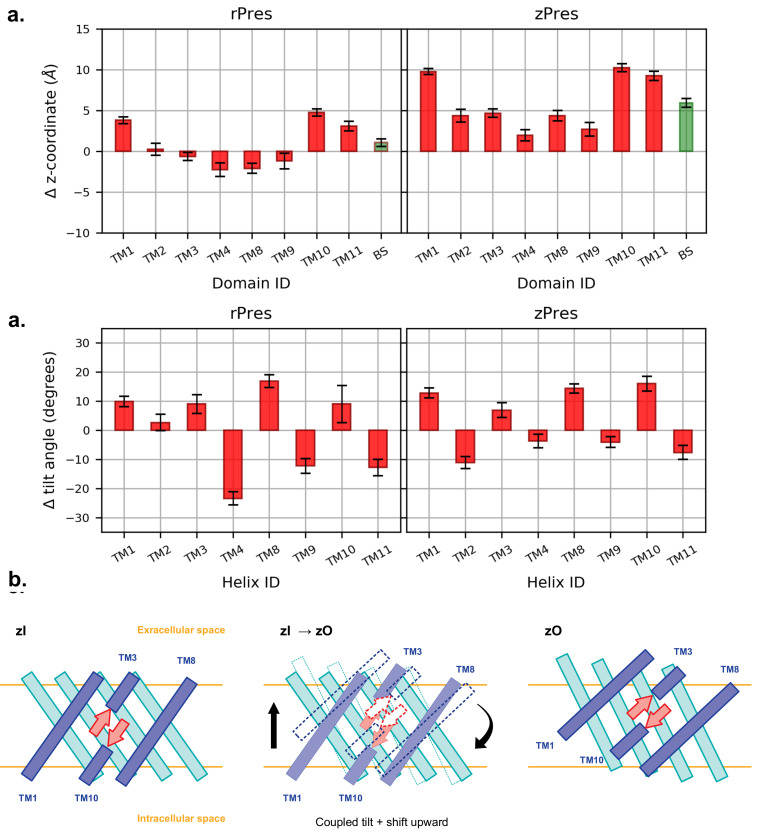
(**a**) Average shift along the *z*-axis of the center of mass of each core TM in the outward state (O), with respect to the most representative inward conformation (I), after structural alignment of the gate. BS indicates the β-sheet. A positive value indicate an upward movement of the center of mass when going from I to O. (**b**) Average difference in the tilt angle of the core TMs, computed comparing each frame of the O simulations and the representative I structures. The tilt angle is taken as the angle between the helix axis and the *z*-axis. (**c**) Relative disposition of the core TMs belonging to the inner shell (blue) and outer shell (cyan) and of the β-sheet forming the binding cavity (red), in zI and zO conformations. The latter is the result of a shift and a tilt, which together preserve the secondary structure of the β-sheet and lift it towards the extracellular side.

**Figure 6 ijms-22-08318-f006:**
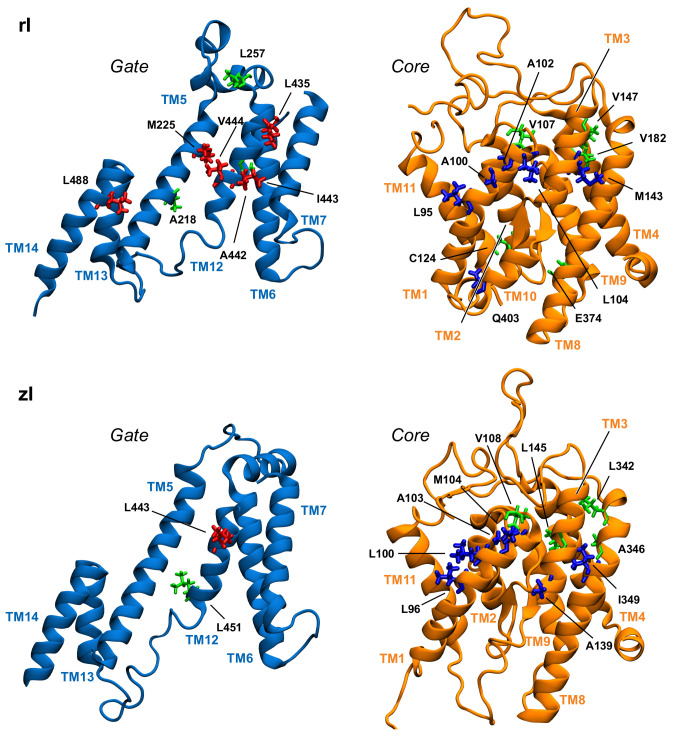
Position in the gate and core domains of the network hubs (in licorice) in the inward-open simulations, as viewed from the interdomain interface. Residues on gate and core that are located at the interface are colored red and blue, respectively. Hub residues not located at the interface are colored green.

**Figure 7 ijms-22-08318-f007:**
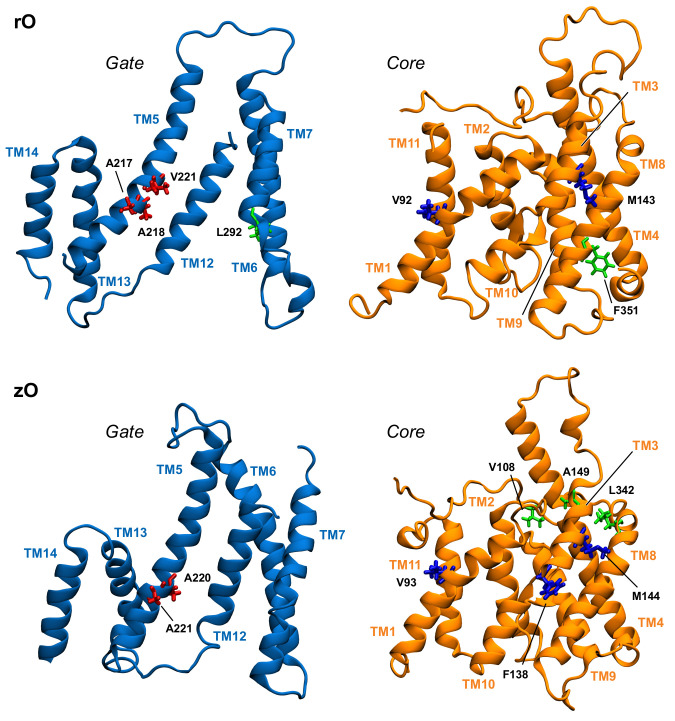
Position in the structure of gate and core domains of the network hubs (in licorice) in the outward-open simulations, as viewed from the interdomain interface. Residues on gate and core that are located at the interface are colored red and blue, respectively. Hub residues not located at the interface are colored green.

**Figure 8 ijms-22-08318-f008:**
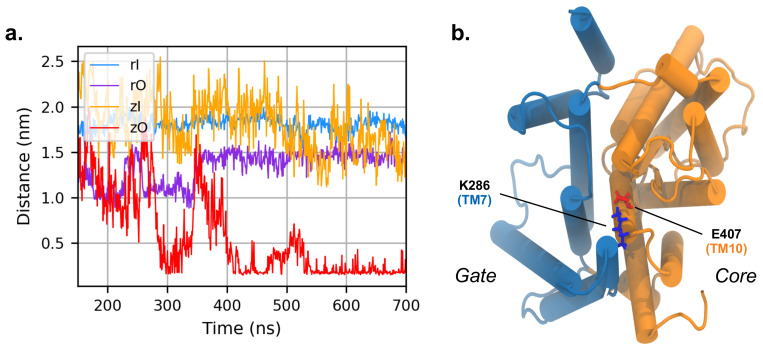
(**a**) Distance between K286 and E407 during the simulation of zPres and rPres (in the latter case, the corresponding residues are K283 and E404). (**b**) Cytoplasmic view of the zO; E407 is in red and K286 in blue. The core and gate domains are highlighted in orange and light blue, respectively.

**Figure 9 ijms-22-08318-f009:**
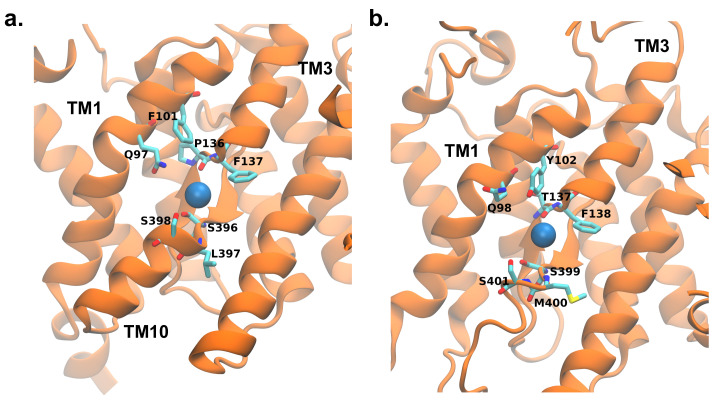
Binding of chloride ions as occurred during the simulation with applied electric field in rI (**a**) and zI (**b**). The highlighted residues are the functional amino acids identified in the work by Walter et al. [9].

**Figure 10 ijms-22-08318-f010:**
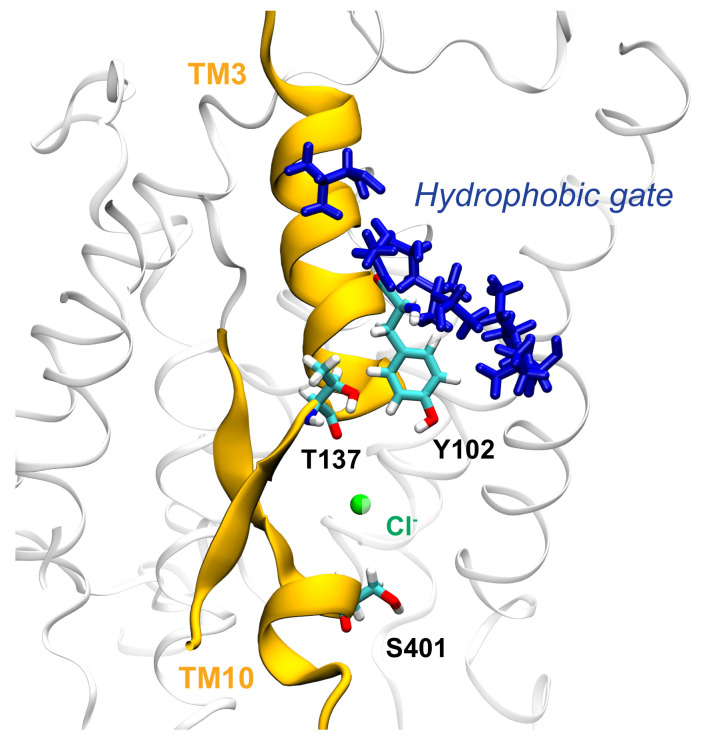
Cl${}^{-}$ ion in the binding cavity of zI where it can interact with the hydroxyl groups of Y102 and T137; we suggest that these strong electrostatic interactions can drive the conformational change, by leading to the opening of the hydrophobic cluster (residues A101, A103, L105, V108, I446, I447) located above the binding site.

**Table 1 ijms-22-08318-t001:** Residues with the largest number of interactions, as computed with PyInteraph [49]. The number of connections (indicated in parentheses for each residue) includes H-bonds, salt bridges, and hydrophobic interactions.

Domain	rPres In	rPres Out	zPres In	zPres Out
Core	L95 (8)	V92 (7)	L100 (8)	V93 (8)
	A100 (7)	M143 (7)	I145 (8)	V108 (7)
	A102 (7)	F351 (7)	A346 (8)	F138 (7)
	L104 (7)		L96 (7)	M144 (7)
	V107 (7)		A103 (7)	A149 (7)
	C124 (7)		M104 (7)	L342 (7)
	M143 (7)		V108 (7)	
	V147 (7)		A139 (7)	
	V182 (7)		L342 (7)	
	E374 (7)		I349 (7)	
	Q403 (7)			
Gate	V444 (8)	A217 (8)	L443 (7)	A220 (7)
	A218 (7)	A218 (8)	L451 (7)	A221 (7)
	M225 (7)	V221 (7)		
	L257 (7)	L292 (7)		
	L435 (7)			
	A442 (7)			
	I443 (7)			
	L488 (7)			

## Data Availability

The data presented in this study are available on request.

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
