# Peer review of "Comparative Molecular Dynamics Investigation of the Electromotile Hearing Protein Prestin"

_ijms, 2021, doi:10.3390/ijms22158318_

Round 1
Reviewer 1 Report
This manuscript is indeed a great addition to the literature and contains multiple figures (mainly in supplementary). The authors investigated the major structural and functional features of a anion transporter Prestin via MD simulation and modeling. However, it’s required major edition of text, graphs and tables because the audience of this paper could be non-computational biologists. I believe more figures can be moved to the main text, as almost 21 figures are at supplementary currently, and that distrusts the reading and deliberation on the results.
Unclear sentences/english editing/editing some figures, and adding details to some parts are required:
- Among other tasks, they are responsible for the passage across the cell membrane of vitamins
or, via the exchange of chloride ions and bicarbonate, for the pH regulation of blood, as
in the case of the band 3 protein
- in the organ of Corti.
- However, the TMD domain of the single protomer is the basic
71 unit for the transport of anions, that is incomplete in mammalian prestin and is the basic
72 process for cell electromotility. Hence we focused only on the TMD domain of the single
- remove coma: simulations, of the interactions
- please specify what type of experimental observations: experimental observations available in literature
- in “The best scored method”, based on what criteria you determined the ‘best’. Please specity.
- please specify your rational for choosing POPC for this D modeling. And add your decription to the text.
Quality of figures: figure 1 and there is no lable
- Caption figure 2 is incomplete and requires ore explanation. Color coding should be also described in the figure so that the reader can easily understand the figure without a need to read the caption. No abbreaviation should be used in figures, it must be as transparent and clear for readers as possible.
- Figure 3: define STATS in the figure caption.
- Table 1: describe what ###:# means. There is always a possibility that the reader of your manuscript are not familiar with MD simulation/modeling but they are interested in your protein target. So better to describe data clearly.
- Please add to the text why “Principal Component Analysis (PCA)” is needed for this analysis.
- “We acknowledge the CINECA award HP10B2Y23Y-ELEVATOR under the ISCRA
528 initiative for the availability of high-performance computing resources.” Should be in acknowledgment section not part of discission/conclusions.
- “We firmly believe that our comparative investigation of prestin represents an addi522
tional step for the understanding of NLC, although further computational studies are
523 desirable to elucidate the possible cooperative mechanisms upon dimer formation, or
524 the interaction between the TMD and STAS domains. Further experimental evidence
525 may shed more light on the proposed mechanism and the key roles pointed out by our
526 computational analysis and critical assessment of the interaction networks.”
I appreciate the authors comment on their analysis and discovery but the text should be a bit unbiassed. I highly suggest the first part “We firmly believe that “ to be removed.
Please explain what experimental data you mean, structural, biochemical assays. Biophysical and by what techniques.
Author Response
We thank reviewer 1 for his/her positive feedback and comments. We have considered all his/her suggestions and amended our manuscript accordingly. The changes are described in the following and are highlighted in the main text.
R: I believe more figures can be moved to the main text, as almost 21 figures are at supplementary currently, and that distrusts the reading and deliberation on the results.
A: We merged figures S11-S12 in one single panel, as well as figures S15-S16; figures S17-S18 were merged in a single panel and promoted to the main text.
R: Unclear sentences/english editing/editing some figures, and adding details to some parts are required:
- Among other tasks, they are responsible for the passage across the cell membrane of vitamins or, via the exchange of chloride ions and bicarbonate, for the pH regulation of blood, as in the case of the band 3 protein
A: This sentence was rephrased and supplemented by additional details and references.
R: 1. in the organ of Corti.
A: The full sentence was rephrased and some details were added to clarify the role of prestin.
R: 2. However, the TMD domain of the single protomer is the basic unit for the transport of anions, that is incomplete in mammalian prestin and is the basic process for cell electromotility. Hence we focused only on the TMD domain of the single
A: This sentence was rephrased and supplemented by additional references.
R: 3. remove coma: simulations, of the interactions
A: This sentence was rephrased.
R: 4. please specify what type of experimental observations: experimental observations available in literature
A: We added references to the types of experimental data we used for comparison with our results.
R: 5. in “The best scored method”, based on what criteria you determined the ‘best’. Please specity.
A: This sentence was rephrased and missing details were added.
R: 6. please specify your rational for choosing POPC for this D modeling. And add your decription to the text.
A: We described the rationale behind our choice of the POPC membrane.
R: Quality of figures: figure 1 and there is no lable
A: The quality of Figure 1 was improved and the labels added; the caption was also rephrased.
R: 8. Caption figure 2 is incomplete and requires ore explanation. Color coding should be also described in the figure so that the reader can easily understand the figure without a need to read the caption. No abbreaviation should be used in figures, it must be as transparent and clear for readers as possible.
A: The color coding was added to the figure, the abbreviations were extended and the caption was rephrased.
R: 9. Figure 3: define STATS in the figure caption.
A: We added this information in the caption of Figure 3.
R: 10. Table 1: describe what ###:# means. There is always a possibility that the reader of your manuscript are not familiar with MD simulation/modeling but they are interested in your protein target. So better to describe data clearly.
A: The caption of the table was rephrased, to make it more clear and detailed.
R: 11. Please add to the text why “Principal Component Analysis (PCA)” is needed for this analysis.
A: We thought that the aim of our PCA analysis was more clearly described in the Results section 3.4; however, the suggestion by the reviewer might improve the readability of the Methods section. Accordingly, we decided to stress the reason for the PCA analysis also in the Methods section by adding the following sentence: “Principal component analysis (PCA) was performed on all simulations of the inward-open states, for both rPres and zPres, in order to compare the exploration of the conformational space in presence and in absence of the electric field.”
R: 12. “We acknowledge the CINECA award HP10B2Y23Y-ELEVATOR under the ISCRA initiative for the availability of high-performance computing resources.” Should be in acknowledgment section not part of discission/conclusions.
A: We moved this part to the “Acknowledgements” section.
R: 13. “We firmly believe that our comparative investigation of prestin represents an additional step for the understanding of NLC, although further computational studies are desirable to elucidate the possible cooperative mechanisms upon dimer formation, or the interaction between the TMD and STAS domains. Further experimental evidence may shed more light on the proposed mechanism and the key roles pointed out by our computational analysis and critical assessment of the interaction networks.”
I appreciate the authors comment on their analysis and discovery but the text should be a bit unbiassed. I highly suggest the first part “We firmly believe that “ to be removed.
A: We removed the expression “We firmly believe”
R: Please explain what experimental data you mean, structural, biochemical assays. Biophysical and by what techniques.
A: We added details on the experiments that we expect might be able to improve our current understanding of the functioning mechanism of prestin, in relation to both its non-linear capacitance and electromotility.
Reviewer 2 Report
The manuscript by Abrusci and colleagues provide some structural insights into putative action mechanisms of two homologous proteins, rat and zebrafish prestins, obtained by homology modeling and molecular dynamics simulations. Since the mammalian homolog lacks the ability to transport chemicals and functions as a motor protein instead, the results would allow for hypothesizing what are molecular determinants required for the transition from complete to incomplete transporter. I believe that the paper can be published upon the addressing the following points:
- Why did the authors not use the recent structure of closely related human SLC26A9 (PDB code 7CH1) as a template for homology modeling?
- From Fig. 2, it’s clear that the secondary structure of both proteins in both states undergo sufficient changes (e.g., a short helix between TM6 and TM7 disappears in rO). Could the authors perform a detailed analysis of the secondary structure?
- Given that one of the major points of the manuscript is to compare mammal and non-mammal homologs of prestin, the authors should provide a detailed analysis of the conservativity of structurally and physiologically relevant residues in both proteins. A thorough analysis comprising the conservativity of amino acid positions among mammalian and non-mammalian homologs should include (but not limited to) the chloride ion binding site: is the putative site proposed here conservative between mammals and non-mammals? The same should be done for the identified clusters of (putatively) physiologically relevant residues.
- The authors mention that some mutations in prestin may lead to its functional impairment. A more comprehensive overview of such mutations from the structural perspective would be highly beneficial for this study. The authors may rely upon the gnomAD [https://doi.org/10.1038/s41586-020-2308-7] and similar databases.
Author Response
We thank Reviewer 2 for his/her positive comments and suggestions. We reply here to each issue:
R: 1. Why did the authors not use the recent structure of closely related human SLC26A9 (PDB code 7CH1) as a template for homology modeling?
A: As explained in section 3.1, SLC26Dg was preferred to SLC26A9 as a template for the proteins in the inward-open state because we considered it as a better model from a functional point of view: we remark in the manuscript that SLC26Dg has been reported to function as a typical transporter, while there is no clear evidence on whether SLC26A9 behaves like a channel or a transporter. Moreover, a large collection of experimental data is available on the SLC26Dg, in particular regarding the function of its transmembrane domain and its response to ion binding, which is the main focus of this manuscript. More recent investigations on the SLC26A9 protein, although certainly of relevance for the overall system, did not add much to the working mechanism of the transport function of the single transmembrane domains, and were mainly focused on the dimerization process and on the role played by the STAS domain and the N-terminal domain; these aspects were not at the core of our present study. However, our models were compared to the experimental structure of SLC26A9, showing a good agreement; the results of this comparison are present in the Supplementary Material.
We added a reference to the experimental structure of human SLC26A9 in the manuscript; the whole paragraph has been rephrased, to better explain the rationale behind the choice of this template.
R: 2. From Fig. 2, it’s clear that the secondary structure of both proteins in both states undergo sufficient changes (e.g., a short helix between TM6 and TM7 disappears in rO). Could the authors perform a detailed analysis of the secondary structure?
A: We added plots with the secondary structure analyses in the Supplementary Material. We added a few comments in the main text, in Section 3.2.
R: 3. Given that one of the major points of the manuscript is to compare mammal and non-mammal homologs of prestin, the authors should provide a detailed analysis of the conservativity of structurally and physiologically relevant residues in both proteins. A thorough analysis comprising the conservativity of amino acid positions among mammalian and non-mammalian homologs should include (but not limited to) the chloride ion binding site: is the putative site proposed here conservative between mammals and non-mammals? The same should be done for the identified clusters of (putatively) physiologically relevant residues.
A: Thorough analyses of the conservativity of prestin residues are already present in literature, both as comparison among sequences of SLC26 homologues (including prestin [1]) and among mammalian and non mammalian prestin sequences (including rat and zebrafish ones [2]). In the manuscript, we explicitly refer to these studies when we report the conservativity of those residues that appear to play a major role for the interaction with the substrate from our simulations. We also discussed the high conservativity of a specific segment in the rPres sequence, and formulated hypotheses on its functional role.
To clarify these issues, we added a figure with the alignment of rPres and zPres sequences, and stressed the references to the sequence comparisons that are already present in literature.
R: 4. The authors mention that some mutations in prestin may lead to its functional impairment. A more comprehensive overview of such mutations from the structural perspective would be highly beneficial for this study. The authors may rely upon the gnomAD [https://doi.org/10.1038/s41586-020-2308-7] and similar databases.
A: We thank the reviewer for pointing out this database, of which we were not aware. However, a search with the SLC26A5 did not result in any indication that could be related to the present study. The reason is probably the fact that we are focusing on a comparison between rat and zebrafish prestin and these may differ from human homologs. However, any discussion on the functional impairment of prestin should take into account the fact that the function of prestin involves at least two intertwined but distinct phenomena: non-linear capacitance and electromotility. Therefore, we preferred to rely on the comparison between the results from our computational study and the experimental mutagenesis data already present in literature, since the latter clearly distinguish the specific structural and functional relevance of the individual mutations. We would also like to point out that comprehensive overviews of prestin mutations are already widely present in literature [3-10].
[1] Chi, X., Jin, X., Chen, Y., Lu, X., Tu, X., Li, X., Zhang, Y., Lei, J., Huang, J., Huang, Z. and Zhou, Q., 2020. Structural insights into the gating mechanism of human SLC26A9 mediated by its C-terminal sequence. Cell discovery, 6(1), pp.1-10.
[2] Liu, Z., Li, G.H., Huang, J.F., Murphy, R.W. and Shi, P., 2012. Hearing aid for vertebrates via multiple episodic adaptive events on prestin genes. Molecular biology and evolution, 29(9), pp.2187-2198.
[3] Rapp, C., Bai, X. and Reithmeier, R.A., 2017. Molecular analysis of human solute carrier SLC26 anion transporter disease-causing mutations using 3-dimensional homology modeling. Biochimica et Biophysica Acta (BBA)-Biomembranes, 1859(12), pp.2420-2434.
[4] Walter, J.D., Sawicka, M. and Dutzler, R., 2019. Cryo-EM structures and functional characterization of murine Slc26a9 reveal mechanism of uncoupled chloride transport. Elife, 8, p.e46986.
[5] Chang, Y.N., Jaumann, E.A., Reichel, K., Hartmann, J., Oliver, D., Hummer, G., Joseph, B. and Geertsma, E.R., 2019. Structural basis for functional interactions in dimers of SLC26 transporters. Nature communications, 10(1), pp.1-10.
[6] Schaechinger, T.J., Gorbunov, D., Halaszovich, C.R., Moser, T., Kügler, S., Fakler, B. and Oliver, D., 2011. A synthetic prestin reveals protein domains and molecular operation of outer hair cell piezoelectricity. The EMBO journal, 30(14), pp.2793-2804.
[7] Gorbunov, D., Sturlese, M., Nies, F., Kluge, M., Bellanda, M., Battistutta, R. and Oliver, D., 2014. Molecular architecture and the structural basis for anion interaction in prestin and SLC26 transporters. Nature communications, 5(1), pp.1-13.
[8] Geertsma, E.R., Chang, Y.N., Shaik, F.R., Neldner, Y., Pardon, E., Steyaert, J. and Dutzler, R., 2015. Structure of a prokaryotic fumarate transporter reveals the architecture of the SLC26 family. Nature structural & molecular biology, 22(10), pp.803-808.
[9] Lovas, S., He, D.Z., Liu, H., Tang, J., Pecka, J.L., Hatfield, M.P. and Beisel, K.W., 2015. Glutamate transporter homolog-based model predicts that anion-π interaction is the mechanism for the voltage-dependent response of prestin. Journal of Biological Chemistry, 290(40), pp.24326-24339.
[10] Drew, D. and Boudker, O., 2016. Shared molecular mechanisms of membrane transporters. Annual review of biochemistry, 85, pp.543-572.
Round 2
Reviewer 2 Report
The manuscript can be published in its present form.